# Modeling driver behavior in the dilemma zone based on stochastic model predictive control

**Wenjun Li[1], Lidong Tan[1]\*, Ciyun Lin**[1,2]

1 Department of Traffic and Transportation, Jilin University, Changchun, China, 2 Jilin Engineering Research Center for ITS, Changchun, China

\* tanld@jlu.edu.cn

## Abstract

Driver behavior is considered one of the most important factors in the genesis of dilemma zones and the safety of driver-vehicle-environment systems. An accurate driver behavior model can improve the traffic signal control efficiency and decrease traffic accidents in signalized intersections. This paper uses a mathematical modeling method to study driver behavior in a dilemma zone based on stochastic model predictive control (SMPC), along with considering the dynamic characteristics of human cognition and execution, aiming to provide a feasible solution for modeling driver behavior more accurately and potentially improving the understanding of driver-vehicle-environment systems in dilemma zones. This paper explores the modeling framework of driver behavior, including the perception module, decision-making module, and operation module. The perception module is proposed to stimulate the ability to perceive uncertainty and select attention in the dilemma zone. An SMPC-based driver control modeling method is proposed to stimulate decision-making behavior in the dilemma zone. The operation module is proposed to stimulate the execution ability of the driver. Finally, CarSim, the well-known vehicle dynamics analysis software package, is used to verify the proposed models of this paper. The simulation results show that the SMPC-based driver behavior model can effectively and accurately reflect the vehicle motion and dynamics under driving in the dilemma zone.

## 1 Introduction

When a driver approaches a signalized intersection as the green signal turns yellow, he or she will face a decision-making or hesitation situation, namely, whether to decelerate to stop before the stop-line or to accelerate to go through the intersection. In this situation, some of the drivers who are approaching the signalized intersection will find themselves too close to the intersection to stop comfortably and safely or too far from the stop-line to pass through within the legal speed limit. The statistical data from the government authorities show that this situation accounts for nearly 50% of traffic accidents that happened at a signalized intersection in China [1–3]. This situation relates to car speed, driver reaction time and decision-making, the geometric parameters of the road and intersection, the duration of the yellow signal light, etc. In the 1960s, Denos and Rober [4] first stated this problem, used kinetic equations to

**Funding:** The authors acknowledge the National Natural Science Foundation of China (Grant No. 51408257), and the Science and technology

project of Jilin Provincial Education Department (Grant No. JJKH20170810KJ) are partly support this work. The awards is received by Ciyun Lin (LCY) The funders had no role in study design, data collection and analysis, decision to publish, or preparation of the manuscript.

**Competing interests:** The authors have declared that no competing interests exist.

analyze the influences on this problem, and tried to optimize the duration of the yellow light to avoid this problem. Olson and Rothery [5] calculated the minimum safe stopping distance and maximum critical crossing distance under different speeds, perception/reaction times and acceleration/deceleration rates to optimize the yellow signal for different geometric dimensions of signalized intersections with kinetic equations. Thirty-five years later, Denos and Rober [6] confirmed that the dilemma zone is an example of the incompatibility of man-made law and physically attainable human behavior. They tried to propose a yellow signal duration consistent with the geometry of a signalized intersection that considered broader ranges of vehicle characteristics to eliminate the conflict. However, they found that it is difficult to eliminate the dilemma zone situation regardless of how slowly the vehicle is moving in any yellow signal duration. Until now, researchers are still interesting in dilemma zone identification to enhance the safety at signalized intersection for drivers by implementing various safety measures [7–9].

Driver behavior is considered one of the most important factors in the genesis of dilemma zones and the safety of human vehicles in this situation. Driver decisions in dilemma zones could result in crash-prone situations at signalized intersections, as an improper decision to stop by the leading driver, combined with the following driver deciding to go, can result in a rear-end collision. Sahar and Montasir [10] proposed a novel safety surrogate measure to capture the degree and frequency of dilemma zone-related conflicts at each approaching intersection. Papaioannou [11] analyzed the relationship between the dilemma zone and the safety level of signalized intersections. This research indicated that a large percentage of drivers facing the yellow signal are caught in the dilemma zone due to high approaching speeds and exercise aggressive behavior. More than half of the drivers choose to cross the STOP line instead of stopping, showing that drivers are neither afraid of the law nor believe that an accident may be caused as a consequence of their choice. Gates [12] and Kim [13] evaluated the stopping characteristics of vehicles in the dilemma zone using video cameras. The field study showed that the different types of drivers and vehicles had different responses in the dilemma zone. Long [14] used the fuzzy decision tree model to analyze the driver decision to go or stop in the dilemma zone at a signalized intersection, considering the vehicle location, velocity and remaining time of the yellow signal. Qi [15] converted the traffic signal light and drivers into a double game model, and through quantification of their earnings under different choice conditions, determined the optimum the driver decision-making via the Nash equilibrium solution concept. Mohammed and Arash [16] applied well-known artificial intelligence algorithms to predict driver stop/run decisions at the onset of a yellow indication for different roadway surface conditions. The research results showed that the driver aggressiveness parameter can be estimated by monitoring the driver historical response to yellow indications. Savolainen, Sharma and Gates [17, 18] investigated how signal timing strategies such as yellow signal duration, all-red clearance interval, advance warning flasher, and automated camera enforcement impact driver decision-making. Lu [19] used high-resolution event-based data to analyze the yellow-light running behavior of drivers. The research showed that snowing weather conditions cause more yellow-light running events. Bar-Gera and Musicant [20] used naturalistic data from digital enforcement cameras to quantify yellow signal driver behavior. The results showed that the frequencies of entrance time after yellow onset are relatively stable during the beginning of the yellow phase. The duration of frequency reduction from 90% to 10% varies considerably across the signalized intersection, and the entrance time ranges from 1.9 s to 2.9 s after a yellow onset indication. Dong [21], Li [22, 23], and Najmi [24] analyzed the influence factors of go/stop decision-making at onset of yellow at signalized intersections, and modeled driving behaviors to formulate strategies to reduce unsafe driving in dilemma zone.

Different drivers exhibit their respective traffic behavior, which is affected by internal factors and external factors. The internal factors are the driver self-attributes, including driver gender, age, driving experiences, cultural background, etc. The external factors are the driving environment, including traffic conditions, road surface conditions, and weather conditions. It is difficult to use kinetic equations to model traffic behavior in controlling vehicles when the drivers approach signalized intersections [25]. However, driver behavior can be partly reflected by the changes in vehicle motion before, during, and after decision-making and after vehicle operation. The driver perceives surrounding information, estimates the driving situation, makes decisions to control the vehicle, and then changes the vehicle motion status by adjusting the steering wheel or changing the vehicle speed by stepping on the gas or brake pedal. These processes can be indicated by high-fidelity vehicle dynamics models that consider the driver to have the internal abilities of driving considering perception, traffic condition estimation, and driving experimental learning and the external abilities of operating vehicles under various driving scenarios. Compared with the normal vehicle kinetic model, the vehicle dynamics model must be more adaptive, stochastic, and time-invariant. These kinds of models can represent driver behavior [26, 27]. Yang [28], Cairano [29] and Dominic [30] pointed out that driver behavior has stochastic characteristics. Driver behavior can be modeled and explained with stochastic variables or stochastic processes. Therefore, when a driver drives to a signalized intersection at the onset of a yellow signal indication, the driver behavior can be described as [31]: (1) The driver perceives the traffic environment to generate the expected driving state. (2) The driver estimates the driving state based on driving experience and current driving states. (3) The driver operates the vehicle (such as accelerating to go through or decelerating to stop) to make the estimated driving state keep pace with the expected driving state. (4) The driver persistently estimates and optimizes the driving state to achieve the expected driving state. Throughout the driving processes, driver behavior is affected by the road surface conditions, traffic flow conditions, weather conditions and driver physiology. In the whole process, driver behavior is consistent with the characteristics and principle of model predictive control (MPC) [32–34]. However, although MPC can deal with disturbances and uncertainties, it is based on the min-max method and cannot efficiently model driver behavior in the dilemma zone, which needs to consider the disturbance and uncertainty of a person-vehicle-road system in real time at a signalized intersection. Stochastic model predictive control (SMPC) has the advantage of analyzing and modeling the system, which is stochastic and uncertain [29]. In this paper, we attempt to use SMPC to model driver behavior in the dilemma zone. The aims of this paper are as follows.

1. A mathematical modeling method is used to study driver behavior in the dilemma zone based on stochastic model predictive control, along with considering the dynamic characteristics of human cognition and execution.

2. Providing a feasible solution for modeling driver behavior more accurately and then potentially improving the understanding of the driver-vehicle-environment system in the dilemma zone, exploring the modeling framework of driver behavior in the dilemma zone, including the perception module, decision-making module, and operation module.

3. We propose an SMPC-based high-fidelity vehicle dynamics and motion model to describe the processes of driver behavior at the onset of yellow signal indication when approaching the signalized intersection and use the CarSim simulator to verify the validity of the proposed model.

The rest of the paper is structured as follows: Section 2 presents the model of driver behavior in the dilemma zone. The simulation and model verification are presented in Section 3.

Section 4 concludes this article with a summary of contributions and limitations, as well as perspectives on future work.

## 2 Model driver behavior in the dilemma zone

### 2.1 Model framework

When a driver approaches the signalized intersection, he or she will perceive the traffic environment, including the traffic light, the distance from the stop-line and the traffic flow in the approaching direction. Then, he or she will estimate the traffic environment and driving state (vehicle motion status) to decide whether to stop before the stop-line or to pass through the signalized intersection. After the decision is made, he or she optimizes the driving operation and operates the vehicle based on his or her driving experience, reaction time, age, ability to handle vehicle, etc. Therefore, the whole process can be divided into three modules: the perception module, the decision-making module, and the operation module. The framework of the driving behavior is shown in Fig 1.

### 2.2 Perception module

When driving to the signalized intersection at the onset of a yellow signal indication, the driver perceives traffic conditions by sensory organs such as vision, auditory, and tactile. All traffic condition information is processed, conducted, and interpreted by the driver brain. Then, the brain generates the expected driving state, which is the ability to predict the vehicle motion status based on the driver experience and current vehicle motion status [35]. Therefore, the perception module model is composed of a traffic condition perception model and an expectation driving state model.

**2.2.1 Traffic condition perception.** The driver perceives road surface conditions, traffic conditions, and vehicle interior motion status by sensory organs. However, not all the information obtained from sensory organs is processed by the brain. The information is selected, processed and comprehended by the cognitive neural system of the driver [36, 37]. In different driving environments, the objects concerning the drivers are different. The driver pays more attention to the objects that impact the driving status most. Regarding selective attention, Broadbent [38] considered that there is a large amount of information stimulation from the surroundings, but the ability of the sensory channel to receive information and the ability of

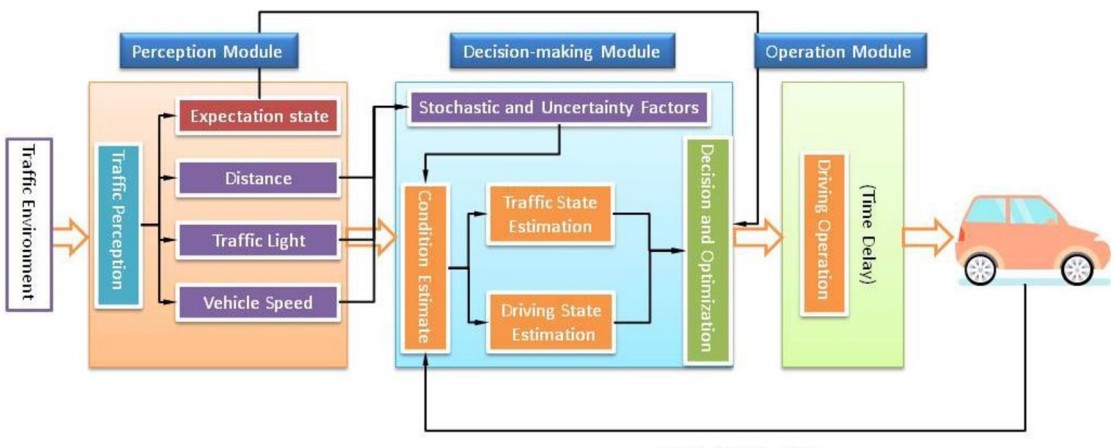

**Fig 1. The framework of driving behavior in the dilemma zone.**

the cognitive system to process information are limited. Therefore, it is necessary to filter and adjust the large amount of information input from the outside world. There is only one channel that passes through a filter into the advanced analysis phase, and this filter demonstrates the selectivity of attention. Then, Deutsch proposed the response selection model (RSM) in 1963 [39]. Compared to the filter model (FM) proposed by Broadbent in 1958, Deutsch considered that there are several channels that pass through the filter and enter the advanced analysis phase. Kahneman [40] proposed the capacity division model (CDM) in 1973. Kahneman considered that selective attention is essentially a resource allocation mechanism that allocates the limited information processing capacity of human beings according to the resource allocation scheme under the constraints of various factors. Based on the theory proposed by Broadbent, Deutsch and Kahneman, researchers have proposed many hypotheses such as the maximum stimulus hypothesis, minimal stimulus hypothesis, and capability of the maximum stimulus number hypothesis to analyze the critical threshold of the filter [41]. However, the selective attention models mentioned above are based on the theory of psychology. In driver behavior modeling, we do not focus on quantizing the complex psychological characteristics of drivers and use only the research results of selective attention that consider the complex psychological characteristics of humans and affect drivers in different driving scenes.

In this paper, we use the theory of RSM and the maximum stimulus hypothesis to describe driver selective attention in the dilemma zone as follows: When a driver approaches the signalized intersection at the onset of the yellow signal indication, the objects in the driving environment will stimulate the driver sensory organ [42]. When the stimulus exceeds a certain threshold value, the driver will keep his/her mind on the objects in the driving environment [43, 44]. This phenomenon is expressed by the following mathematical formula:

$$SA(t) \in \{i | ST_i(t) > ST_c\} \tag{1}$$

where SA(t) is the set of selected attention objects in the driving environment at time t. i is the $i^{th}$ object in the driving environment. $ST_i(t)$ is the $i^{th}$ object stimulus to the driver at time t in the driving environment. $ST_c$ is the threshold of stimulus to the driver to receive attention in the driving environment.

When a driver is caught in the dilemma zone, he or she will pay more attention to the traffic light, current vehicle distance from the stop-line, and current vehicle speed, which are the key factors for the driver to make decisions in the proceeding driving operation in the dilemma zone [45]. In the driver behavior modeling, we consider only the ability of driving surrounding perception affecting drivers to make decisions in dilemma zones. Therefore, the effect of traffic condition perception on driver decision-making in the dilemma zone is described as AF{SA(t)}. According to the references of [46, 47], AF{SA(t)} depends on the driver psychophysiological characteristics and the time of the driver confronting the yellow signal in the dilemma zone. Therefore, we assume that AF{SA(t)} is a random variable subordinate to a specific statistical property. In different prediction periods, the effects of driver traffic condition perception AF{SA(t)} are independent.

**2.2.2 Expectation driving state.** After perceiving the driving surroundings in the dilemma zone, the driver will generate an expected driving state in mind before deciding whether to pass through the signalized intersection or not. The expected driving state relies on the vehicle current speed, the distance from the forward vehicle or from the stop-line, and the rest time of the yellow signal. In this process, the driver focuses on the vehicle location and vehicle speed. Therefore, we can describe that the expected driving state as the change in location when the vehicle approaches the signalized intersection.

$$S_{\upsilon_1}(t+1), S_{\upsilon_2}(t+2), \cdots, S_{\upsilon_{N_E}}(t+N_E) \tag{2}$$

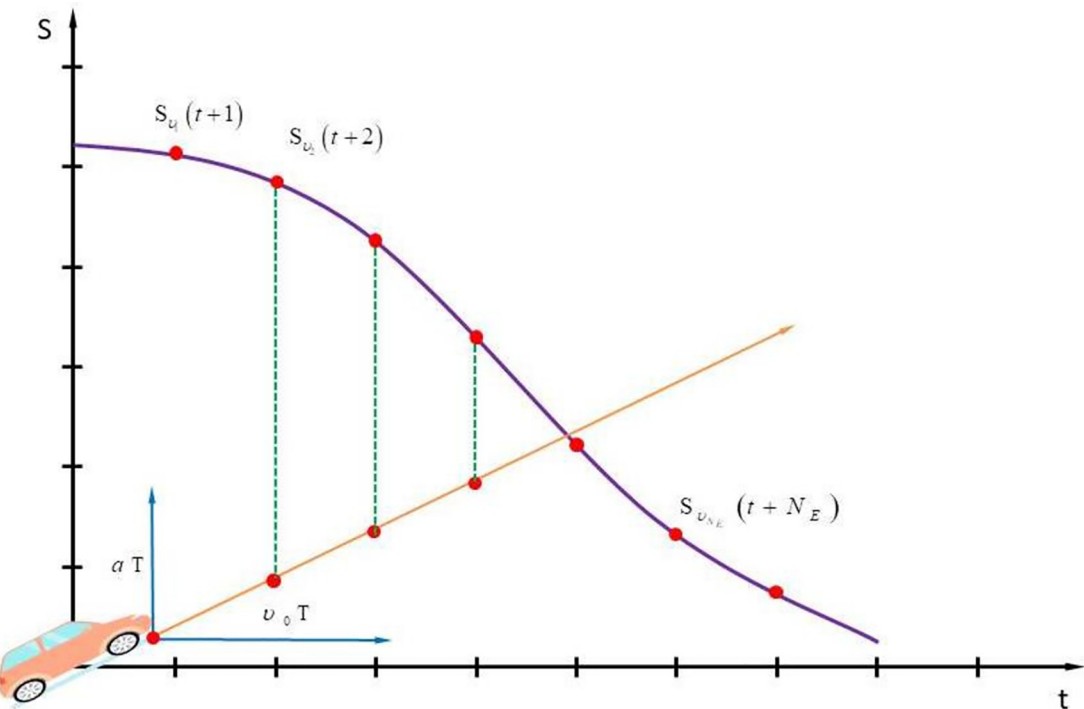

**Fig 2. Diagram of the expected driving state in the dilemma zone.**

where $S_{\upsilon 1}(t + 1)$, $S_{\upsilon 2}(t + 2)$, and $S_{\upsilon N_E}(t + N_E)$ are the expected vehicle distances from the stop line at times $t + 1$, $t + 2$, and $t + N_E$, respectively. t is the time the driver realizes the dilemma zone or the time start to expect the driving state. $\upsilon_1$, $\upsilon_2$, and $\upsilon_{N_E}$ are the vehicle speeds at times t + 1, t + 2, and t + $N_E$, respectively. $N_E$ relates to the degree of driver expectation. A greater value of $N_E$ means more concentration in the driving environment and a better traffic condition for the approaching entrance lane of the signalized intersection.

When a driver approaches the signalized intersection at the onset of a yellow signal indication, the initial speed of the vehicle during the perception-expectation-response period T is defined as $\upsilon_0$, and the acceleration of the vehicle in the period is defined as $\alpha$. As the vehicle location and the yellow signal status change, the expectation driving state of the driver will change in every reaction period. The diagram of the expected driving state in the dilemma zone is shown in Fig 2.

## 2.3 Decision-making module

When the driver realizes that he or she is caught in the dilemma zone with real-time selective attention to the driving environment, he or she must decide whether to decelerate to stop before the stop-line or to accelerate to pass through the signalized intersection. In this situation, the driver evaluates the driving conditions, makes a decision, and optimizes the driving state to cope with the decision and to drive safely and comfortably. Therefore, we divide the decision-making module into two models: (1) the driving condition estimation model, and (2) the decision-making and optimization model.

**2.3.1 Driving condition estimation.** In the process of decision making in the dilemma zone, the driver will estimate the forward traffic state of the road and the driving state of the vehicle. To describe these characteristics, we try to model the driver cognitive characteristics

in this situation [48, 49].

$$\begin{cases} M\dot{v}_y + \sigma \cdot M v_x = F_y f + F_x \delta \\ I_z \dot{\sigma} = \xi \cdot F_y f - \zeta \cdot F_x \delta \end{cases} \tag{3}$$

where $M$ is the quality of vehicle; $v_y$ and $v_x$ are the lateral and transverse speed of vehicle, respectively; $F_y$ and $F_x$ are the driver selective attention to the forward traffic state, and around traffic state of the vehicle which are reflected by the lateral and transverse braking force, respectively; $f$ and $\delta$ are the response time of driver lateral and transverse braking force, respectively; $I_z$ is the rotational inertia, used to deal with the case where driver changes the driving lane, accelerating and overtaking to pass through the signalized intersection; $\sigma$ is the angular speed; $\xi$ and $\zeta$ are the lateral and transverse distance from braking force to the centroid of the vehicle.

The lateral and transverse braking forces $F_y$ and $F_x$ are closely related to the impact factors of selective attention AF{SA(t)}, loading and vehicle capacity. We rewrite $F_y$ and $F_x$ as:

$$\begin{cases} F_y f = C_f \cdot \mu \cdot a_f \\ F_x \delta = C_\delta \cdot \mu \cdot a_\delta \end{cases} \tag{4}$$

where $\mu$ is used to describe the effect of selective attention in the dilemma zone during driving; $a_f$ and $a_\delta$ are the lateral and transverse accelerations under the braking force, respectively; and $C_f$ and $C_\delta$ are the lateral and transverse capacities of the vehicle, respectively.

Putting formula (4) into formula (3), formula (3) can be rewritten as:

$$\begin{cases} \dot{v}_y = \dfrac{\mu(C_f \cdot a_f + C_\delta \cdot a_\delta)}{M} - \sigma \cdot v_x \\ \dot{\sigma} = \mu(\xi \cdot C_f \cdot a_f - \zeta \cdot C_\delta \cdot a_\delta)/I_z \end{cases} \tag{5}$$

With the kinematic equation of vehicle motion, the vehicle movement can be described as:

$$\begin{cases} \dot{x}(t) = v_x cos\varphi(t) - v_y(t) sin\varphi(t) \\ \dot{y}(t) = v_x sin\varphi(t) + v_y(t) cos\varphi(t) \end{cases} \tag{6}$$

where $x(t)$ and $y(t)$ are the lateral and transverse displacements of the vehicle, respectively, and $\varphi$ is the yaw angle of the vehicle heading.

In the dilemma zone, the foremost reaction of the driver is to consider decelerating to stop before the stop-line or accelerating to go through the intersection. The yaw angle in this paper is considered close to zero. The formula (6) can be rewritten as follows:

$$\begin{cases} \dot{x}(t) = v_x - v_y(t)\varphi(t) \\ \dot{y}(t) = v_x \varphi(t) + v_y(t) \end{cases} \tag{7}$$

Combining formula (5) and formula (7), the state of driving condition estimation can be formed as a fourth-order status variable:

$$X(t) = \begin{pmatrix} v_y(t) \\ \sigma(t) \\ y(t) \\ \varphi(t) \end{pmatrix} \tag{8}$$

Formula (5) and formula (7) can be combined as a matrix:

$$
\begin{cases}
\dot{X}(t) = A(\mu)X(t) + B(\mu)\tau(t) \\
Y(t) = CX(t)
\end{cases}
\tag{9}
$$

where $A(\mu) = \begin{pmatrix} -\frac{(C_f+C_\delta)\mu}{Mv_x} & -\frac{(\xi \cdot C_f - \zeta \cdot C_\delta)\mu}{Mv_x} - v_x & 0 & 0 \\ -\frac{(\xi \cdot C_f - \zeta \cdot C_\delta)\mu}{I_z v_x} & -\frac{(\xi^2 \cdot C_f + \zeta^2 \cdot C_\delta)\mu}{I_z v_x} & 0 & 0 \\ 1 & 0 & 0 & v_x \\ 0 & 1 & 0 & 0 \end{pmatrix}$ and $B(\mu) = \begin{pmatrix} \frac{C_f \mu}{M} \\ \frac{\xi \cdot C_f \mu}{I_z} \\ 0 \\ 0 \end{pmatrix}$ are the status

variable coefficient matrixes; $\tau(t)$ is the coefficient of correction, which is used to represent the stochastic and uncertainty factors; and $C$ is the constant vector, $C = (0\ 0\ 1\ 0)$.

In every perception-expectation-response period T, the dynamic driving condition estimation can be described as a discretization equation.

$$
X(t+1) = A(\mu(t))X(t) + B(\mu(t))\tau(t)
\tag{10}
$$

**2.3.2 Decision-making and optimization.** After estimating the traffic condition and vehicle state while making decisions in the dilemma zone, the driver will evaluate the vehicle motion and optimize the vehicle status in the next reaction period. We try to use the dynamic discretization equation to describe the estimated state based on stochastic model prediction control theory [33, 34, 50].

$$
\begin{pmatrix} X(t+1) \\ X(t+2) \\ \vdots \\ X(t+N_E) \end{pmatrix}
$$

$$
= \begin{pmatrix} A(\mu(t)) \\ A^2(\mu(t)) \\ \vdots \\ A^{N_E}(\mu(t)) \end{pmatrix} X(t)
$$

$$
+ \begin{pmatrix} B(\mu(t)) & \cdots & 0 \\ A(\mu(t))B(\mu(t)) & \cdots & 0 \\ \vdots & \vdots & \vdots \\ A^{N_E-1}(\mu(t))B(\mu(t)) & \cdots & \sum_{i=1}^{N_E} A^{i-1}(\mu(t))B(\mu(t)) \end{pmatrix} \begin{pmatrix} \tau(t) \\ \tau(t+1) \\ \vdots \\ \tau(t+N_E-1) \end{pmatrix}
\tag{11}
$$

$$
\begin{pmatrix} Y(t+1) \\ Y(t+2) \\ \vdots \\ Y(t+N_E) \end{pmatrix} = \begin{pmatrix} C & 0 & \cdots & 0 \\ 0 & C & \cdots & 0 \\ \vdots & \vdots & \cdots & \vdots \\ 0 & 0 & \cdots & C \end{pmatrix} \begin{pmatrix} X(t+1) \\ X(t+2) \\ \vdots \\ X(t+N_E) \end{pmatrix}
\tag{12}
$$

The evaluation driving status can be formulated as:

$$
DS(t) = E(\mu(t))X(t) + F(\mu(t))R(t)
\tag{13}
$$

where $DS(t)$ is the vector of estimation approaching vehicle status in the dilemma zone at

time t; $E(\mu(t)) = \begin{pmatrix} CA(\mu(t)) \\ CA^2(\mu(t)) \\ \vdots \\ CA^{N_E}(\mu(t)) \end{pmatrix}$;

$$F(\mu(t)) = \begin{pmatrix} CB(\mu(t)) & 0 & \cdots & 0 \\ A(\mu(t))B(\mu(t)) & CB(\mu(t)) & \cdots & 0 \\ \vdots & \vdots & \vdots & \vdots \\ A^{N_E-1}(\mu(t))B(\mu(t)) & A^{N_E-2}(\mu(t))B(\mu(t)) & \cdots & \sum_{i=1}^{N_E} A^{i-1}(\mu(t))B(\mu(t)) \end{pmatrix}$$; and

$$R(t) = \begin{pmatrix} \tau(t) \\ \tau(t+1) \\ \vdots \\ \tau(t+N_E-1) \end{pmatrix}.$$

As a driver obtains the vehicle motion status by estimation in mind, he or she tries to optimize the vehicle status to make the driving comfortable and safe.

$$J(t) = \sum_{i=1}^{N_E} \|DS(t+i) - O_p(t+i)\|^2 + \sum_{i=1}^{N_E} \|\tau(t+i-1) - \rho(t+i-1)\|^2 \tag{14}$$

where $J(t)$ is the object of driver decision, that is, to stop before the stop-line or pass through the signalized intersection; $O_p(t)$ is the optimized operation of the driver at time t to obtain the final object; and $\rho(t)$ is the random disturbance in the decision or operation in the dilemma zone at time t.

As $O_p(t)$ is considered based on the traffic condition perception and driving state expectation, it can be rewritten as:

$$O_p(t+i) = S_\upsilon(t+i)con\varphi(t) + iTv_x sin\varphi(t) + y(t) \tag{15}$$

## 2.4 Operation module

Due to the restrictions of driver driving experiences, response ability, and cognitive characteristics, the action of optimization and the expected vehicle state will be delayed. Delays may occur in the process of traffic perception, decision-making, driving expectation and optimization. We use the delay transfer function to describe this phenomenon as [25]:

$$DT(t) = \sum_{i=1}^{n} e^{-t_d(i)} \tag{16}$$

where $DT(t)$ is the total delay time of the process from traffic perception to vehicle operation in the dilemma zone, and $t_d(i)$ is the delay parameter in the $i^{th}$ section during the process.

Based on the stochastic model prediction control theory, we can obtain the driver reaction and operation in every step in the dilemma zone and model the rolling horizon driver behavior model in the dilemma zone by using the sequence formulas in the perception module, decision-making module and operation module.

## 3 Simulation and model verification

In this paper, we use the CarSim 8.02 simulator (Mechanical Simulation Corporation, Ann Arbor, Michigan, United States) to verify the validation of the proposed model in this paper based on SMPC. CarSim is universally the preferred tool for analyzing vehicle dynamics, developing active controllers, calculating the performance characteristics of a car, and engineering next-generation active safety systems. CarSim includes configurable high-fidelity and real-time vehicle dynamics models, which can accurately reflect vehicle dynamics and motion under various driving conditions, and includes models of complex road and road structures, high-fidelity traffic, weather and lighting conditions, and vehicle models for cars, SUVs, trucks, and buses [51]. In China, cars and SUVs make up the bulk of traffic flow in urban city. The Hatchback and SUV models in CarSim are closest to the actual vehicles running on the road. Therefore, we use the vehicle models of A-Class (Hatchback), B-Class (Hatchback), and D-Class (SUV) in the CarSim software package to verify the proposed model in this paper. The vehicle model parameter settings in CarSim are shown in Table 1.

The driver behavior and vehicle dynamics model in the dilemma zone proposed in this paper are defined by formulas and variables added with VS commands and VS configurable functions, handled by built-in controllers and solvers. Traffic light is defined as a target object and combined with ranging sensors in CarSim. Roads and reference paths are created with Scene Builder and defined with configurable functions to build dilemma zone scenarios and traffic event sequences. A snapshot of the traffic scene in the simulator is shown in Fig 3.

In traffic event sequences, we use kinetic equations to define the range of the dilemma zone and assume that:

**Table 1. Vehicle model parameter settings in CarSim.**

| Vehicle Model | M (kg) | $I_z$ (kg/m$^2$) | $C_f$ (N) | $C_\delta$ (N) |
|---|---|---|---|---|
| A-Class (Hatchback) | 747.00 | 288.00 | 8614 | 13649 |
| B-Class (Hatchback) | 1111.00 | 288.00 | 8614 | 13649 |
| D-Class (SUV) | 1429.00 | 377.10 | 9335 | 10541 |

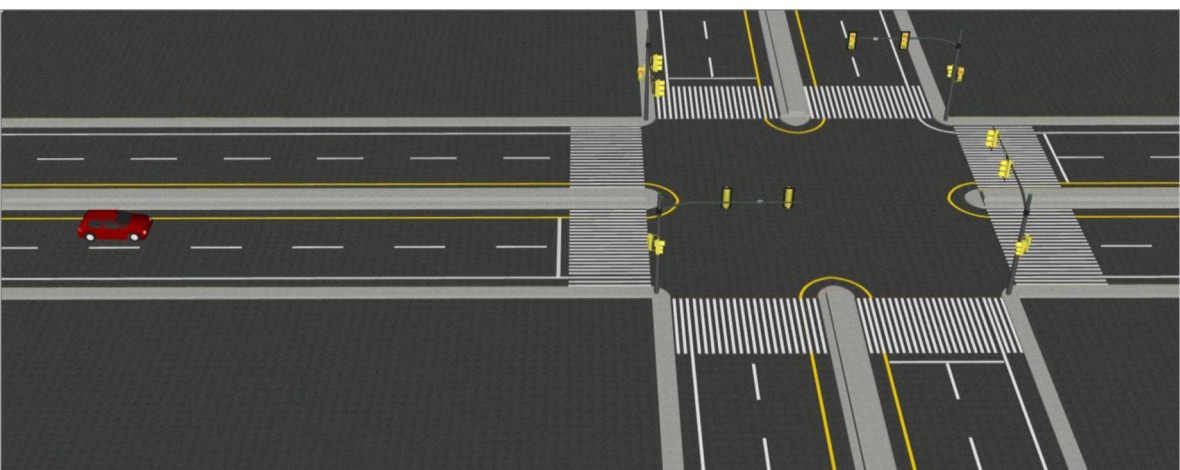

**Fig 3. The snapshot of traffic scene.**

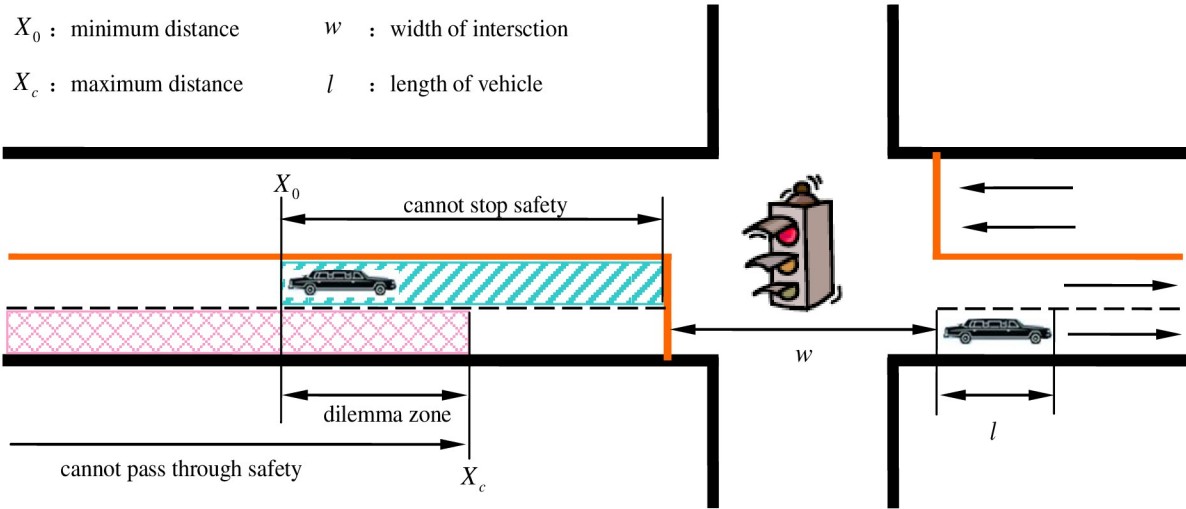

**Fig 4. Type I dilemma zone.**

1. When the driver realizes the yellow signal indication, the initial vehicle speed is $V_0$. If the driver decides to stop before the stop-line, the minimum distance to stop safely is [6]:

$$X_0 = V_0 \cdot \delta + \frac{V_0^2}{2\alpha_-} \tag{17}$$

where $\delta$ is the reaction time delay, and $\alpha_-$ is the vehicle deceleration.

2. If the driver decides to pass through the signalized intersection, the maximum distance to pass through safely is [6]:

$$X_c = V_0 \cdot \delta + V_0 \cdot (\tau - \delta) + \frac{1}{2}\alpha_+ \cdot (\tau - \delta)^2 - (w + l) \tag{18}$$

where $\tau$ is the rest of the yellow signal time; $\alpha_+$ is the vehicle acceleration; $w$ is the width of the intersection; and $l$ is the length of the vehicle.

If $X_c < X_0$, the location of vehicle $X \in (X_c, X_0)$ or if $X_c > X_0$, and the location of vehicle $X \in (X_0, X_c)$, the vehicle is caught in the dilemma zone, as shown in Figs 4 and 5.

To analyze the degree of attention affecting driver behavior, we set $N_E$ as (5, 10, 15) to examine the relationship by comparing the expectation path to the simulation path under different driving conditions. The simulation experiments are divided into two groups: (1) the driver decides to stop before the stop-line; or (2) the driver decides to pass through the signalized intersection. The vehicle speed in the simulation is set as 45 km/h. The range and location of dilemma zone for different driver is different. To make the trajectory data compareable, the observation range is setting as 100 meter along the driving approaching. The starting observation location is set at 100 meter before the stop-line for the vehicles which decide to stop before the stop-line (group 1). For vehicles which decide to go through the signalized intersection, the starting observation location is set at 50 meter before the stop-line (group 2). Running the CarSim software using A-Class (Hatchback), B-Class (Hatchback), and D-Class (SUV) with $N_E = 5$, $N_E = 10$, $N_E = 15$ in group 1 and group 2, respectively, we obtain the pot data of the simulation results and form the comparison diagram as in Figs 6–9.

For group 1, the value deceleration rate begin to change around the location of 50m before the stop-line. It can conclude that the dilemma zone is begin around at 50m before the stop-

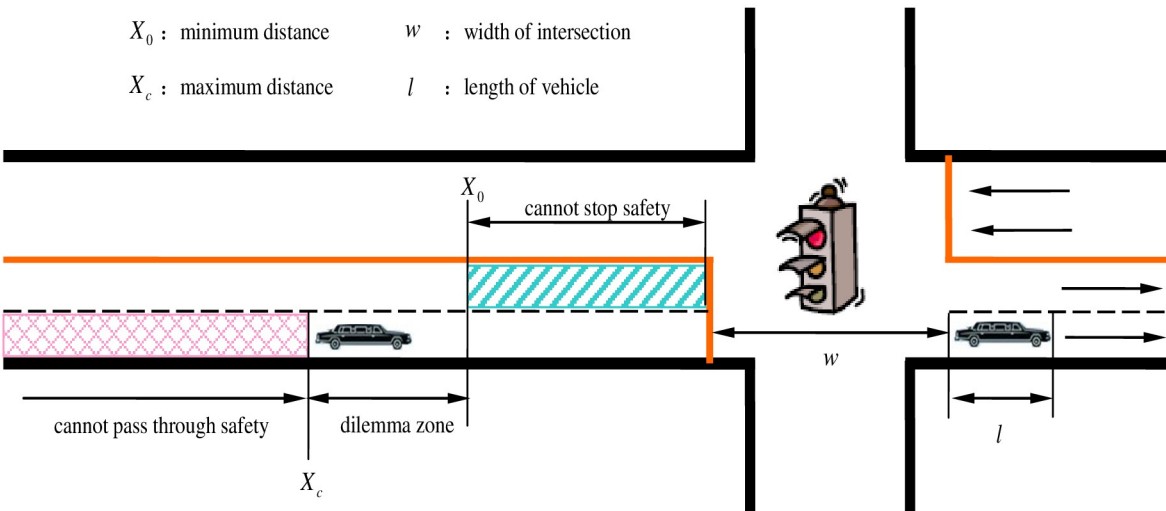

**Fig 5. Type II dilemma zone.**

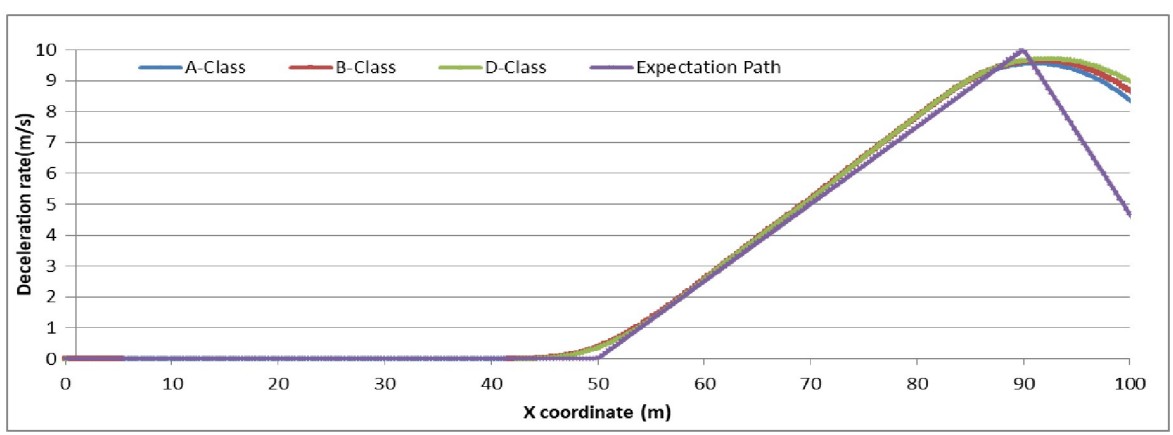

**Fig 6. The vehicle driving trajectory of different vehicle models in $N_E$ = 10 (Group 1).**

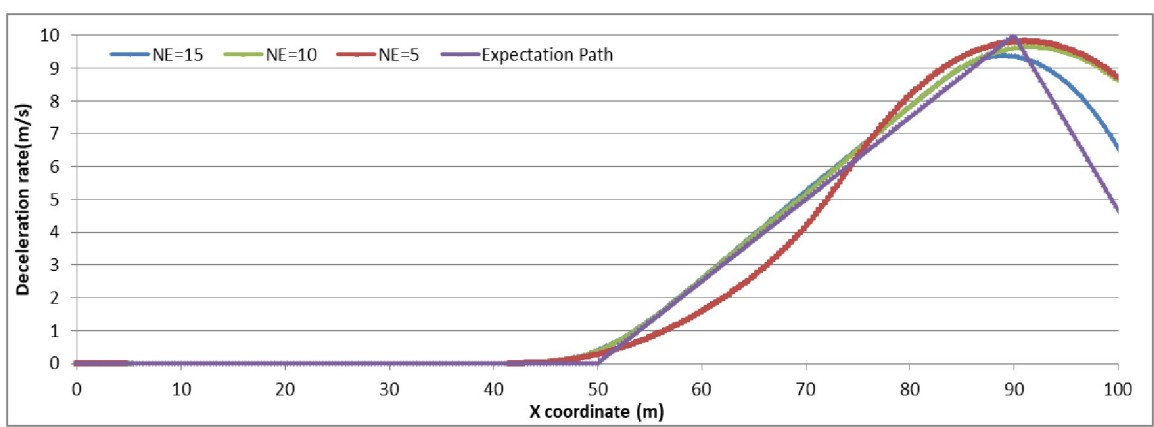

**Fig 7. The B-Class vehicle driving trajectory for different $N_E$ (Group 1).**

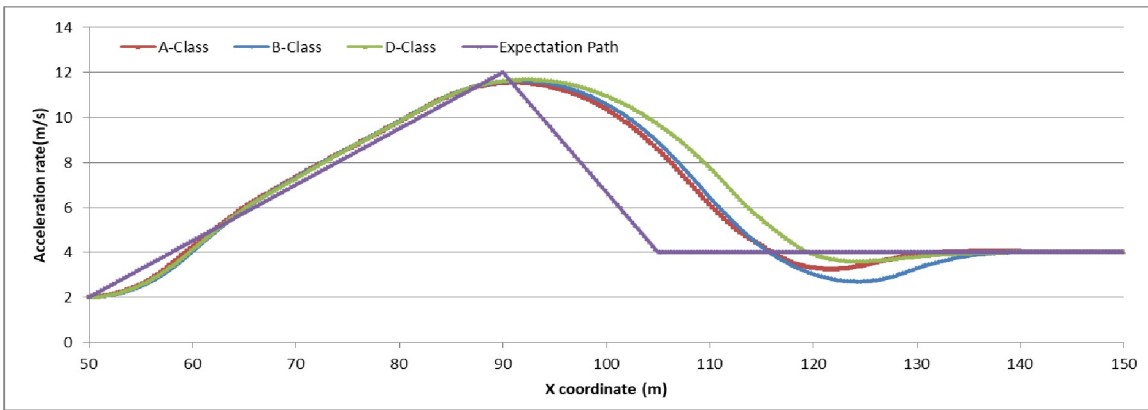

**Fig 8. The vehicle driving trajectory of different vehicle models in $N_E = 10$ (Group 2).**

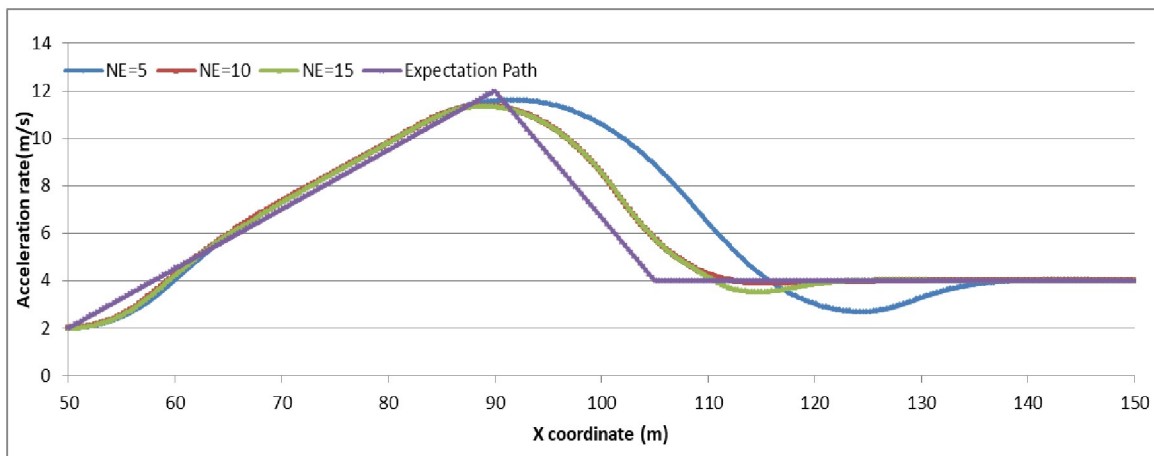

**Fig 9. The B-Class vehicle driving trajectory for different $N_E$ (Group 2).**

line for the vehicle driving at 45km/h of initial velocity. Higher value of $N_E$ means driver palys more attention on traffic condition when driving. From the trajectory data, the less value of $N_E$, the fluctuation of deceleration rate is more pronounced and deviates the expection of driver greater. $N_E = 10$ is enough for driver to keep decelerating the vehicle safety. Furethermore, it can conclude that if the drivers concentrate enough on the traffic condition when driving, the trajectories of vehicle are similarity, and consist with the expectation of driver.

For group 2, the value acceleration rate increases continuously around the location of 50m before the stop-line. It can identify that the dilemma zone forming around 50m before the stop-line for the vehicle driving at 45km/h of initial velocity. It can conclude that if the drivers concentrate enough on the traffic condition when driving, the trajectories of vehicle are similarity, and consist with the expectation of driver. However, when passing through the signalized intersection, the trajectories deviation shows out in different vehicle model, especially in the driver with lower $N_E$ value.

The simulation results and Figs 6–9 show that the more concentrated the driving environment is, the easier it is to obtain the desired path. Based on Figs 6 and 8, the trajectories of

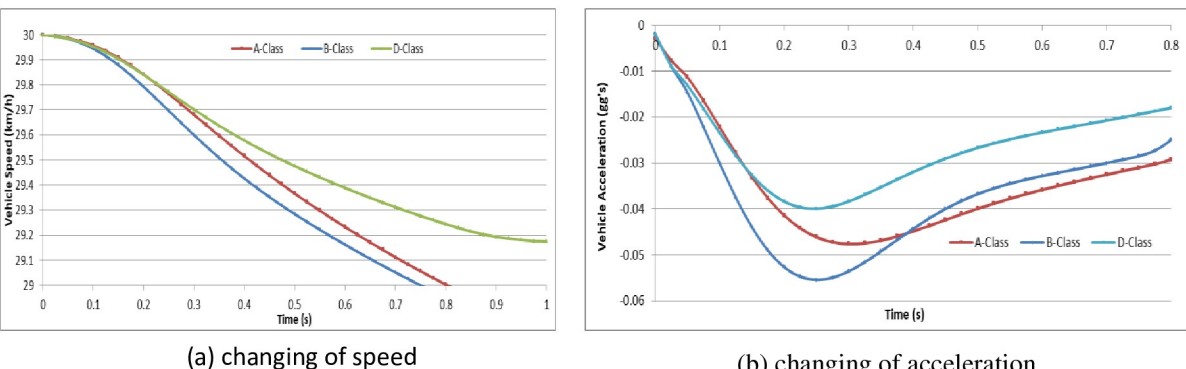

(a) changing of speed

(b) changing of acceleration

**Fig 10. The changes in vehicle speed and acceleration during the reactions at initial speed = 30 km/h with different vehicle models (Group 1).**

A-Class, B-Class, and D-Class with the same driving decision are close to each other. We can conclude that the vehicle models do not obviously affect the driver attention in the dilemma zone in the simulation. Based on Figs 7 and 9, the deviation of vehicle trajectories in different $N_E$ with the same driving decision are not statistically significant at the 95% confidence level. Therefore, the driver attention does not distinctly affect the driver to make the decision to stop before the stop-line or to pass through the signalized intersection.

To analyze the vehicle speed effect on the driver behavior in the dilemma zone, we set the initial speed as 30 km/h, 45 km/h and 70 km/h in the A-Class (Hatchback), B-Class (Hatchback), and D-Class (SUV) vehicle models, respectively [52]. The simulation experiments are divided into two groups: (1) the driver decides to stop before the stop-line; or (2) the driver decides to pass through the signalized intersection. The changes in vehicle speeds and vehicle accelerations during the reactions in the dilemma zone are shown in Figs 10–15.

For group 1, it shows out that the driver stepped on the brake pedal hastily first and then gradually smooth. The higher initial speed of vehicle, the driver will step on the brake pedal deeper and more haste. The lower initial speed of vehicle, the driver has more time to stop the vehicle smoothly before the stop-line. In addition, the different vehicle models show out different acceleration performance.

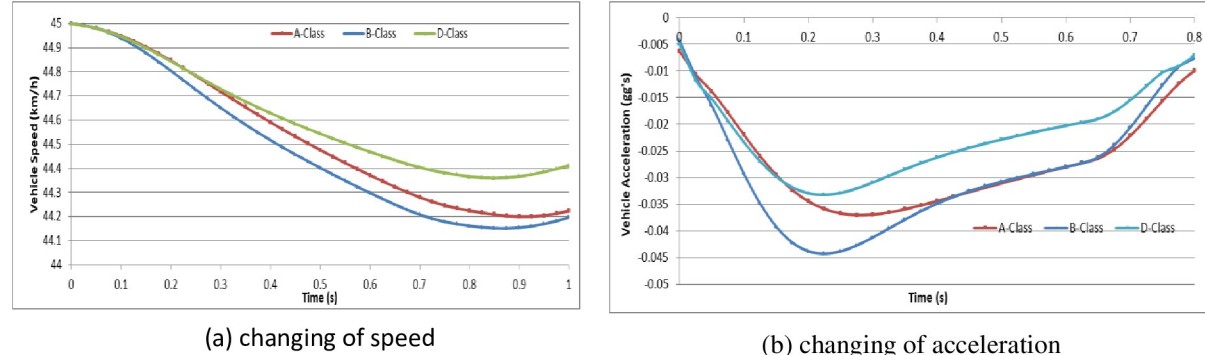

(a) changing of speed

(b) changing of acceleration

**Fig 11. The changes in vehicle speed and acceleration during the reactions at initial speed = 45 km/h with different vehicle models (Group 1).**

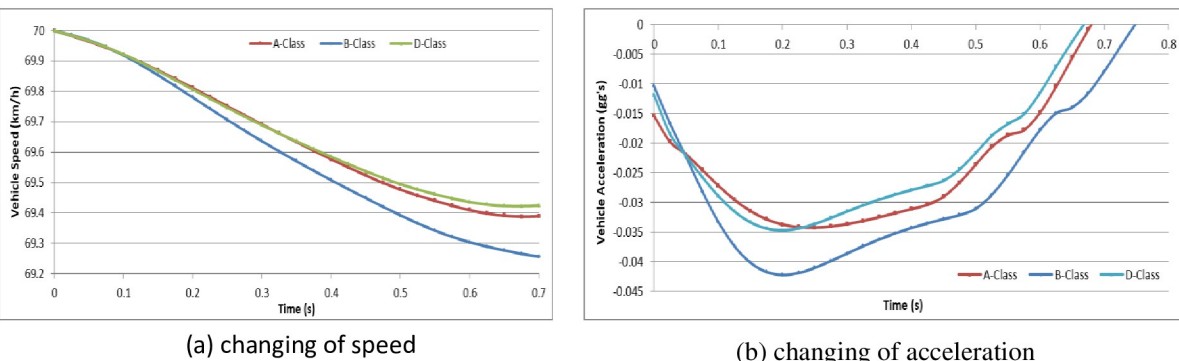

(a) changing of speed

(b) changing of acceleration

**Fig 12. The changes in vehicle speed and acceleration during the reactions at initial speed = 70 km/h in different vehicle models (Group 1).**

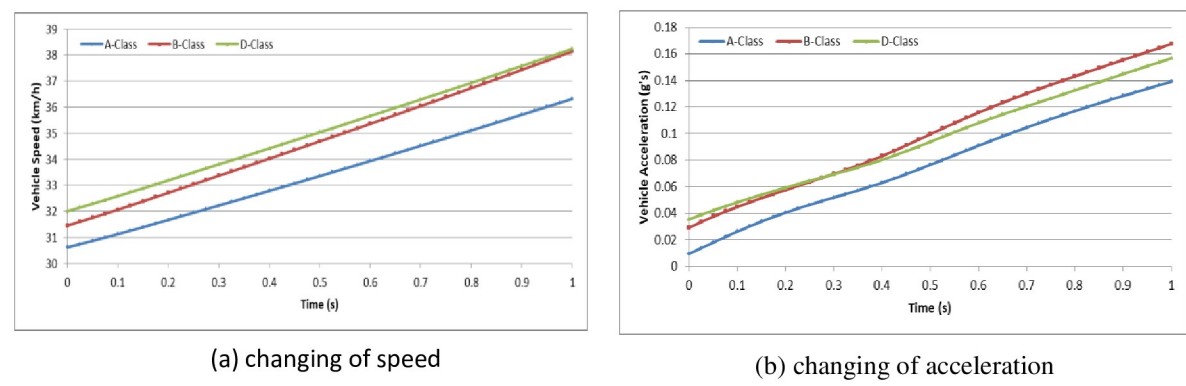

(a) changing of speed

(b) changing of acceleration

**Fig 13. The changes in vehicle speed and acceleration during the reactions at initial speed = 30 km/h with different vehicle models (Group 2).**

For group 2, it shows out that the driver stepped on the gas pedal gradually but with different strength. The lower initial speed of vehicle, the driver will step on the gas pedal deeper. The lower initial speed of vehicle, the driver need more time to accelerate the vehicle to go through the signalized intersection. And the driver will step on the gas pedal deeper to obtain more

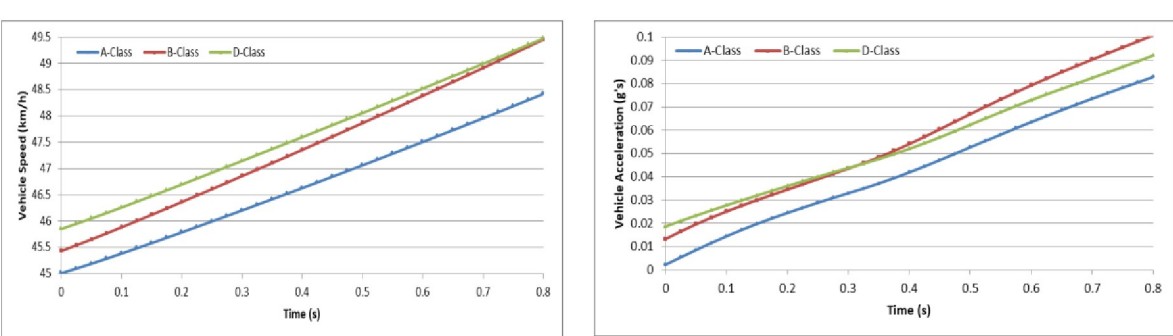

**Fig 14. The changes in vehicle speed and acceleration during the reactions at initial speed = 45 km/h with different vehicle models (Group 2).**

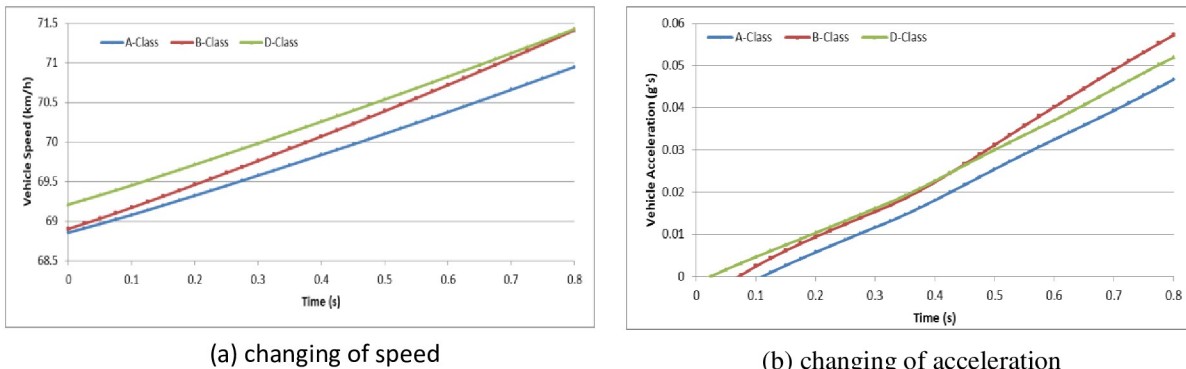

(a) changing of speed                                              (b) changing of acceleration

**Fig 15. The changes in vehicle speed and acceleration during the reactions at initial speed = 70 km/h with different vehicle models (Group 2).**

acceleration performance. However, the changing of speed in different initial speed seems similarity.

In Simulation Group 1, the simulation results and Figs 10–12 show that the B-class vehicle model seems to be more sensitive to the driving environment than the A-Class and D-Class vehicle models at any initial speed. The D-Class vehicle model braking response is smoother than the braking response of the A-Class and B-Class vehicle models. In Simulation Group 2, the simulation results and Figs 13–15 show that the A-Class and D-Class vehicle models seem to have the same variation tendency in vehicle speed and vehicle acceleration during the response. The B-Class vehicle model has a shaper tendency than the A-Class and D-Class vehicle models in the acceleration response.

Finally, we try to analyze the operation delay time effect on driver expectations. We set $t_d$ as (0.10, 0.15, 0.20) to examine the relationship by comparing the expectation path to the simulation path under different driving conditions. The simulation experiments are divided into two groups: (1) the driver decides to stop before the stop-line; or (2) the driver decides to pass through the signalized intersection. The vehicle speeds in the simulation are set as 30 km/h, 45 km/h and 70 km/h. Running the CarSim software using A-Class (Hatchback), B-Class (Hatchback), and D-Class (SUV) with $t_d = 0.10$, $t_d = 0.15$, $t_d = 0.20$ in group 1 and group 2, respectively, we obtain the pot data of the simulation results and form the comparison diagram as in Figs 16–18.

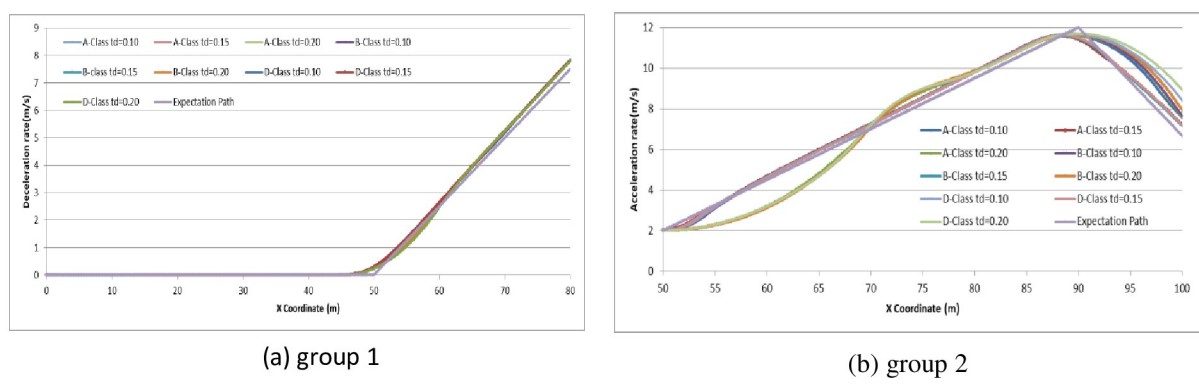

(a) group 1                                              (b) group 2

**Fig 16. Comparison diagram of operation time delay in different vehicle models at initial speed = 30 km/h.**

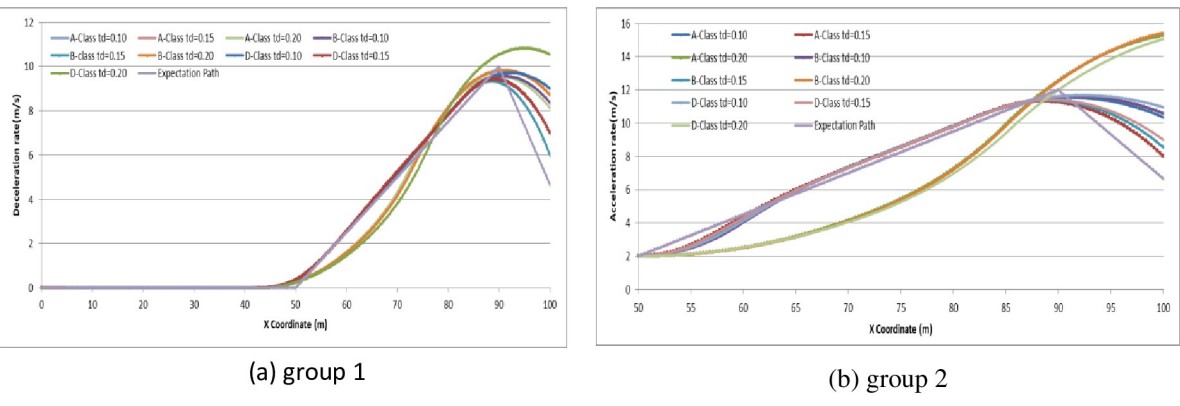

**Fig 17. Comparison diagram of operation time delay in different vehicle models at initial speed = 45 km/h.**

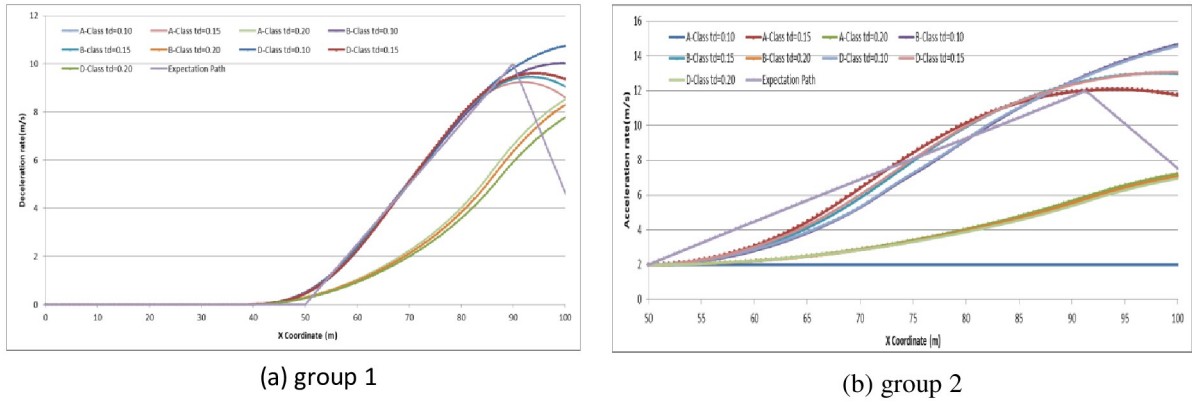

**Fig 18. Comparison diagram of operation time delay in different vehicle models at initial speed = 70 km/h.**

From the simulation results, we can conclude that the longer the operation delay is, the harder it is to follow the expectation path of driver optimization. At an initial speed of 30 km/h, the operation time delay $t_d$ does not significantly influence driving when the driver decides to stop before the stop line. However, when the driver decides to accelerate to pass through the signalized intersection, the more operation time delay $t_d$, the more deviation of expectation will be obtained during driving through. There is no obvious evidence to indicate that the vehicle model affects the operation delay time. As the initial speed reaches 45 km/h and 70 km/h, the deviation of expectation increases and becomes unsteady. The deviation of expectation and unsteadiness is more obvious in the condition of driver deciding to pass through the signalized intersection than to stop before the stop-line. Therefore, when the driver drives at high speed into the dilemma zone, the driver has less operation delay time during driving and decision-making. In other words, the driver needs more concentration on the driving environment to avoid any deviation of driving desired.

## 4 Conclusions

In this paper, we use the stochastic model predictive control method to model driver behavior in dilemma zones. In the driver behavior model, we consider the dynamic characteristics of the driver-vehicle-environment system when the driver approaches the signalized intersection

at the onset of the yellow signal indication. The driver behavior model is divided into three modules: the perception module, decision-making module, and operation module. The perception module is used to model the ability of drivers to perceive traffic conditions and the ability to plan the driving state. The decision-making module is used to model the ability of the driver to realize the dynamic characteristics of vehicle motion and the ability to optimize the vehicle trajectory by evaluating the traffic condition and vehicle driving state. The operation module is used to model the response ability of the driver in the dilemma zone from cognitive to vehicle operation. Based on the SMPC-based high-fidelity dynamics and motion model, the driver behavor can be reflected and estimated by identifying the changes of vehicle motion. In addition, the disturbances and uncertainties casued by internal factors or by external factors during driver approaches the signalized intersction at onset of yellow can be considered by adjust the parameter value of $N_E$ and $t_d(i)$ in dynamics formula.

Finally, we used CarSim, the well-known vehicle dynamics analysis tool, which includes high-fidelity models under various driving scenarios and user-defined built-in capabilities in the software package, to verify the models of this paper. From the simulation results, we can conclude that:

1. If the driver pays more attention to the driving environment, he or she more easily obtains the expectation path. This characteristic is not obvious relative to the vehicle models in which the driver is driving.

2. Driver behavior will be affected by the vehicle model and initial vehicle speed in the dilemma zone. The effect variation tendency seems consistent when the driver decides to accelerate to pass through the signalized intersection, and the extent of variation depends on the vehicle model and initial vehicle speed. If the driver decides to stop before the stop-line, the variation tendency varies according to different vehicle models and initial vehicle speeds.

3. The operation time delay reflects the driver physiological properties. For driving safely in the dilemma zone, if the driver is driving at high speed when approaching the signalized intersection at the onset of the yellow signal indication, more of the driver attention to the driving environment and less operation time delay to obtain the expectation path is required. We may infer that older drivers may tend to drive at lower speeds to obtain the desired driving state in urban cities.

However, some topics remain to be studied, and further research work includes the following: (1) pedestrians and other obstacles in front of driving vehicles, road surface conditions, weather situations and other constraint conditions in dilemma zones should be considered in driver behavior modeling. (2) The field vehicle trajectory data and driver response tests in the dilemma zone should be combined in driver behavior modeling. (3) At present, only simulator-based tests have been performed, and hardware experiments, in-vehicle tests, and scenario tests for the proposed modeling methods should be carried out in the future.

## Supporting information

**S1 Appendix. Source data for simulation and model verification.**
(RAR)

## Acknowledgments

We are very thankful to postgraduate students Yingzhi Guo, Yu zhao, and Ruixin Wei in our research group for their time and efforts in simulation and data collecting. We are very

thankful to the reviewers for their time and efforts. Their comments and suggestions greatly improved the quality of this paper.

## Author Contributions

**Conceptualization:** Wenjun Li.

**Data curation:** Lidong Tan, Ciyun Lin.

**Formal analysis:** Wenjun Li, Ciyun Lin.

**Funding acquisition:** Lidong Tan, Ciyun Lin.

**Methodology:** Wenjun Li, Lidong Tan, Ciyun Lin.

**Validation:** Lidong Tan.

**Writing – original draft:** Ciyun Lin.

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
