## [Decision Letter · Decision Letter 0]

18 Jan 2021

PONE-D-20-38386

Modeling Driver Behavior in the Dilemma Zone Based on Stochastic Model Predictive Control

PLOS ONE

Dear Dr. Lin,

Thank you for submitting your manuscript to PLOS ONE. After careful consideration, we feel that it has merit but does not fully meet PLOS ONE’s publication criteria as it currently stands. Therefore, we invite you to submit a revised version of the manuscript that addresses the points raised during the review process.

We look forward to receiving your revised manuscript.

Kind regards,

Feng Chen

Academic Editor

PLOS ONE

Journal Requirements:

Reviewers' comments:

Reviewer's Responses to Questions

**Comments to the Author**

1. Is the manuscript technically sound, and do the data support the conclusions?

Reviewer #1: Yes

Reviewer #2: Yes

2. Has the statistical analysis been performed appropriately and rigorously? 

Reviewer #1: Yes

Reviewer #2: No

3. Have the authors made all data underlying the findings in their manuscript fully available?

Reviewer #1: Yes

Reviewer #2: Yes

4. Is the manuscript presented in an intelligible fashion and written in standard English?

Reviewer #1: Yes

Reviewer #2: Yes

5. Review Comments to the Author

Reviewer #1: The research proposed a framework for driver behavior analysis in dilemma zone, which could provide the insights to the ability of drivers to perceive traffic conditions and the ability to plan the driving state. The modeling framework of driver behavior includes the perception module, decision-making module, and operation module. The perception module is used to model the ability of drivers to perceive traffic conditions and the ability to plan the driving state. The decision-making module is used to model the ability of the driver to realize the dynamic characteristics of vehicle motion and the ability to optimize the vehicle trajectory by evaluating the traffic condition and vehicle driving state. The operation module is used to model the response ability of the driver in the dilemma zone from cognitive to vehicle operation. The results show that the SMPC-based driver behavior model can effectively and accurately reflect the vehicle motion and dynamics under driving in the dilemma zone. This manuscript has potential of being published at PLOS ONE. I have some comments and suggestions for the authors to consider in their revision.

1. What would be data to calibrate this proposed model of driver behavior? I also would like to see how these types of models are applied in practice.

2. It seems that the model only considers a single vehicle or user. Could it be applied to multiple vehicles, as the intersection, the vehicles are in queen and interaction each other?

3. The dilemma zone in the simulation is fixed zone or dynamic? If the yellow signal adjust according the motion of vehicle when it fall into the dilemma zone?

4. Why only three car models are used in the simulation? To represent the driver behavior model more accurately and then potentially improving the understanding of the driver-vehicle-environment system in the dilemma zone, are more car models needed to be considered?

Reviewer #2: This paper proposed a framework of driver behavior (perception, decision-making, and operation) using a stochastic predictive modeling method.

1. Recommend the authors adding several latest references in literature review.

2. The resolution of Fig. 6-18 needs to be improved.

3. I suggest using real field data to validate at least one of the scenarios from the simulation cases and seeing if the conclusions still valid.

4. Compared with state-of-art prediction methods, what is the strength of the proposed model?

5. The vehicles simulated in this study are only three models. Can more vehicle types be analyzed?

6. Suggest more statistical analysis between the simulated results.

6. PLOS authors have the option to publish the peer review history of their article (what does this mean?). If published, this will include your full peer review and any attached files.

Reviewer #1: No

Reviewer #2: No

---

## [Author Response · Author response to Decision Letter 0]

1 Feb 2021

Dear Reviewers:

Manuscript ID: PONE-D-20-38386

Title: “Modeling Driver Behavior in the Dilemma Zone Based on Stochastic Model Predictive Control”

We wish to express our very deep appreciation, and the appreciation of all of us, to your great efforts and suggestions for our manuscript. They are valuable and very helpful for revising and improving our paper, as well as the important guiding to our researches.

A point-to-point response to your comments are in the "Response to Reviewer".

We tried our best to improve the manuscript and made some changes in the manuscript. These changes will not influence the content and framework of the paper. And here we did not list the changes but marked in red in revised paper. We appreciate for your warm work earnestly, and hope that the correction will meet with approval. Thank you for your time and patience. I look forward to receiving your letter.

Once again, we would like to thank you for the constructive comments and suggestions. Please feel free to contact us with any questions. We are looking forward to your reply.

Yours sincerely,

Authors

---

## [Decision Letter · Decision Letter 1]

8 Feb 2021

Modeling driver behavior in the dilemma zone based on stochastic model predictive control

PONE-D-20-38386R1

Dear Dr. Lin,

We’re pleased to inform you that your manuscript has been judged scientifically suitable for publication and will be formally accepted for publication once it meets all outstanding technical requirements.

Kind regards,

Feng Chen

Academic Editor

PLOS ONE

Additional Editor Comments (optional):

Reviewers' comments:

Reviewer's Responses to Questions

**Comments to the Author**

1. If the authors have adequately addressed your comments raised in a previous round of review and you feel that this manuscript is now acceptable for publication, you may indicate that here to bypass the “Comments to the Author” section, enter your conflict of interest statement in the “Confidential to Editor” section, and submit your "Accept" recommendation.

Reviewer #1: All comments have been addressed

Reviewer #2: All comments have been addressed

2. Is the manuscript technically sound, and do the data support the conclusions?

Reviewer #1: Yes

Reviewer #2: Yes

3. Has the statistical analysis been performed appropriately and rigorously? 

Reviewer #1: Yes

Reviewer #2: Yes

4. Have the authors made all data underlying the findings in their manuscript fully available?

Reviewer #1: Yes

Reviewer #2: Yes

5. Is the manuscript presented in an intelligible fashion and written in standard English?

Reviewer #1: Yes

Reviewer #2: Yes

6. Review Comments to the Author

Reviewer #1: The author has made modifications to reviewing the last version which is much better than the previous one. The statement of contributions has also been more clear. The overall quality of this paper is good, which has theoretical reference value and highlight future directions. In my opinion, it can be accepted.

Reviewer #2: To the authors: All comments have been addressed. Thanks for your explanation. Looking forward to seeing the validation using LiDAR data.

7. PLOS authors have the option to publish the peer review history of their article (what does this mean?). If published, this will include your full peer review and any attached files.

Reviewer #1: No

Reviewer #2: No

---

## [Editor Report · Acceptance letter]

10 Feb 2021

PONE-D-20-38386R1 

Modeling driver behavior in the dilemma zone based on stochastic model predictive control 

Dear Dr. Lin:

I'm pleased to inform you that your manuscript has been deemed suitable for publication in PLOS ONE. Congratulations! Your manuscript is now with our production department. 

Kind regards, 

on behalf of

Dr. Feng Chen 

Academic Editor

PLOS ONE